# Study on the Degradation of a Semi-Synthetic Lignin–Acrylic Acid Hydrogel with Common Bacteria Found in Natural Attenuation Processes

**DOI:** 10.3390/polym15122588

**Published:** 2023-06-06

**Authors:** Humberto D. Jiménez-Torres, Saira L. Hernández-Olmos, Eire Reynaga-Delgado, Eulogio Orozco-Guareño

**Affiliations:** 1Laboratorio de Fisicoquímica del, Departamento de Química, Universidad de Guadalajara, Centro Universitario de Ciencias Exactas e Ingenierías, Blvd. Marcelino García Barragán #1451, Guadalajara 44430, Jalisco, Mexico; 2Departamento de Farmacobiología, Universidad de Guadalajara, Centro Universitario de Ciencias Exactas e Ingenierías, Blvd. Marcelino García Barragán #1451, Guadalajara 44430, Jalisco, Mexico

**Keywords:** biodegradation, hydrogel, bacteria, lignin

## Abstract

In this study, lignin was chemically modified to promote hydrogel degradation as a source of carbon and nitrogen for a bacterial consortium consisting of *P. putida F1*, *B. cereus* and, *B. paramycoides*. A hydrogel was synthesized using acrylic acid (AA), acrylamide (AM), and 2-acrylamido-2-methyl-1-propanesulfonic acid (AMPS) and cross-linked with the modified lignin. The structural changes and mass loss in the hydrogel, as well as its final composition, were evaluated as functions of the growth of the selected strains in a culture broth with the powdered hydrogel. The average loss was 18.4% wt. The hydrogel was characterized using FTIR spectroscopy, scanning electronic microscopy (SEM), elemental analysis (EA), and thermogravimetric analysis (TGA) before and after bacterial treatment. FTIR showed that the carboxylic groups present in both the lignin and the acrylic acid of the hydrogel decreased during bacterial growth. The bacteria showed a preference for the biomaterial components of the hydrogel. SEM demonstrated superficial morphological changes in the hydrogel. The results reveal that the hydrogel was assimilated by the bacterial consortium while preserving the water retention capacity of the material and that the microorganisms carried out a partial biodegradation of the hydrogel. The results of the EA and TGA confirm that the bacterial consortium not only degraded the biopolymer (lignin), but also used the synthetic hydrogel as a carbon source to degrade its polymeric chains and modified original properties. This modification with lignin as a crosslinker (which is a waste product of the paper industry) is therefore proposed to promote hydrogel degradation.

## 1. Introduction

Hydrogels are three-dimensional networks of polymers of natural or synthetic origin with the ability to retain large amounts of water until reaching their physicochemical equilibrium [1].

These materials conserve their shape during swelling due to their cross-linked structure, which is formed with the help of a chemical crosslinker [2].

The synthetic hydrogels with the highest levels of production are those obtained with acrylic acid (AA) and acrylamide (AM) because they are stable and retain large amounts of water [3]. However, after being used for a purpose, hydrogels could contain chemical compounds of which disposal is complicated. Beyond the management of hydrogels, there is the possibility of hydrogels affecting ecosystems when they are used in soil or water. The environmental impact of cross-linked hydrogels remaining in soil or water is poorly understood [4,5,6]. Concerning microorganisms, Wang et al. reported that superabsorbent polymers (SAPs) changed the ecological bacterial diversity in soils by promoting the growth of certain species and reducing the number of others [4]. They also found that when the soil was water-saturated, the SAPs released substances, such as methane sulfonyl chloride, long-chain amides, and esters, which could negatively impact the environment.

Soils and surface aquatic systems contain a great diversity of organisms that have evolved to take up a large variety of polluting compounds as sources of carbon and nitrogen, some of which are of anthropogenic origin. This process is called natural attenuation. For an organism to be able to degrade a compound, this compound must enter one or more metabolic pathways of the organism and thus promote the conversion of the polluting compound into energy, cell mass, and harmless biological by-products [5].

In the case of synthetic hydrogels (SHs), some studies regarding the biological degradability of AA and AM have been performed in recent years [3,6,7]. Although it is known that soil microorganisms can colonize SHs, the general impact of SHs on soil microorganisms remains variable and difficult to evaluate [4,8]. Investigating how bacteria interact with hydrogels to use them as sources of carbon and nitrogen could provide valuable information for future modifications in the selection of more sustainable components, and could further our understanding of the impact of SHs on the microbial biomass of ecosystems.

Certain bacterial groups have stood out for using polluting compounds as carbon sources, particularly the species of the genera *Pseudomonas* and *Bacillus*. *Pseudomonas putida* is found naturally in soil and is characterized by its immense environmental adaptability and biochemical versatility developed through evolution, being able to degrade natural polymers such as lignin, alkanes from C14 to C32, and 61% of the total hydrocarbons of petroleum (TPH), in addition to products such as gasoline, diesel oil, crude oil, and other recalcitrant compounds [9,10,11,12,13].

A semi-synthetic hydrogel was designed and manufactured based on the monomers AA, AM, and 2-acrylamido-2-methyl-1-propanesulfonic acid (AMPS) because they contain functional groups such as –COOH, –OH, –CONH_2_, –CONH, and –SO_3_H. These chemical groups have a high affinity for heavy metals in aqueous solution (Pb, Cr, Cd, and Ni) or chemical compounds that are active pharmaceutical ingredients. To facilitate the hydrogel’s degradation after use, modified lignin was selected as a cross-linker. For this reason, our research group aimed to enable the hydrogel to be used as a carbon source in bacterial metabolic systems.

In this study, we evaluated the modified hydrogel’s ability to maintain the viability of *P. putida F1* and two types of *Bacilli, B. cereus* and *B. paramycoides*, in a bacterial pool where the hydrogel functioned as a source of carbon and nitrogen for these microorganisms. The bacteria were not used individually because the degradation of complex materials is more efficient when microorganisms are used in a consortium. The bacterial growth kinetics were evaluated at a controlled temperature of 35 ± 0.2 °C, without shaking. The hydrogel’s structural changes, mass loss, and final composition were evaluated to analyze its degradation via the degradation of its lignin content and to demonstrate that the bacteria were able to use the carbon chains of the synthetic material as a carbon source.

We used a pool of bacteria rather than individual bacteria in this work, since several authors have reported that plastic biodegradation tests using individual bacteria are less efficient. Shovitri et al. [14] used *Pseudomonas* and *Bacillus* isolated from mangrove waters to degrade single-use plastics using the Winogradsky column method. Then, they demonstrated that the bacteria enacted a bioaugmentation process in marginal soils contaminated with plastics, indicating a positive effect on degradation. However, by using these microorganisms separately, they discussed the effects of the particular enzymes of each microorganism on the plastics, obtaining different results from when the bacteria were used in a consortium. On the other hand, Aravinthan et al. [15] studied the synergistic growth of a *P. azotoformans* and *Bacillus flexus* consortium, obtaining a 22.7% degradation of physically pretreated polypropylene. In previous studies by these authors, it was reported that the effect of using a single bacterium for biodegradation is lower than when using a consortium.

In this study, a pool of bacteria rather than individual bacteria were used in this work, since several authors have reported that plastic biodegradation tests using individual bacteria are less efficient. Shovitri et al. [14] used *Pseudomonas* and *Bacillus* isolated from mangrove waters to degrade single-use plastics using the Winogradsky column method. Then, they demonstrated that the bacteria enacted a bioaugmentation process in marginal soils contaminated with plastics, indicating a positive effect on degradation. However, by using these microorganisms separately, they discussed the effects of the particular enzymes of each microorganism on the plastics, obtaining different results from when the bacteria were used in a consortium. On the other hand, Aravinthan et al. [15] studied the synergistic growth of a *P. azotoformans* and *Bacillus flexus* consortium, obtaining a 22.7% degradation of physically pretreated polypropylene. In previous studies by these authors, it was reported that the effect of using a single bacterium for biodegradation is lower than when using a consortium.

For the above reasons, in this research we evaluated the modified hydrogel’s ability to maintain the viability of *P. putida F1* and two types of Bacilli, *B. cereus* and *B. paramycoides*, in a bacterial pool where the hydrogel functioned as a source of carbon and nitrogen for these microorganisms. The bacteria were not used individually because the degradation of complex materials is more efficient when microorganisms are used in a consortium. The bacterial growth kinetics were evaluated at a controlled temperature of 35 ± 0.2 °C, without shaking. The hydrogel’s structural changes, mass loss, and final composition were evaluated to analyze its degradation via the degradation of its lignin content and to demonstrate that the bacteria were able to use the carbon chains of the synthetic material as a carbon source.

## 2. Materials and Methods

### 2.1. Reagents

Lignin was used as a crosslinking agent, and for this purpose, it was chemically modified to activate at least two functional groups to serve as chemical anchor compounds. Alkali lignin with a low sulfonate content (Sigma-Aldrich, St. Louis, MO, USA), acryloyl chloride (Sigma-Aldrich; St. Louis, MO, USA; purity > 97%), and tetrahydrofuran (THF; Sigma-Aldrich, St. Louis, MO, USA; HPLC grade) were used without further treatment. The monomers 2-acrylamido-2-methyl-1-propanesulfonic acid (AMPS), acrylic acid (AA), and acrylamide (AM) (all obtained from Sigma-Aldrich; St. Louis, MO, USA; purity > 99%) were used to synthesize the hydrogel. Potassium persulfate and sodium bisulfite (both from Golden Bell; Golden, CA, USA; purity > 98%) were used as redox initiators. For the degradation experiments, casein peptone, Muller-Hinton agar (MH), and Luria-Bertani culture broth (LB) were used (all obtained from DIBICO CDMX-México).

### 2.2. Kraft Lignin Modification

The lignin modification was carried out according to the procedure proposed by Rico-Garcia et al. [16]. A solution was prepared by adding 2.5 g of lignin with a low sulfonate content to 25 mL of distilled water and stirring at 1000 ppm until dissolution. Then, a 5 mL aliquot of this solution was transferred to a test tube in an ice water bath at a temperature of 0–4 °C, continuing the stirring for 3 min at 900 rpm. Subsequently, 15 drops of diluted acryloyl chloride were added to the solution dropwise at time intervals of 5 s, and the mixture was left stirring for another 30 min. THF was then added to the test tube, sealed, and vigorously shaken until a homogeneous solution was obtained. A precipitate was then formed and separated from the suspension via decantation, and it was placed on a Petri dish and allowed to rest for 24 h. The solid that formed was placed in an oven at 50 °C for 24 h. Figure 1 shows the proposed reaction between lignin and acryloyl chloride to obtain the modified lignin to be used as a crosslinker.

### 2.3. Hydrogel Synthesis

The synthesis of the hydrogel was performed via free-radical solution polymerization, using potassium persulfate and sodium bisulfite as redox initiators and modified lignin as a crosslinking agent. Amounts of 0.78 g of AM, 4.72 g of AA, 4.52 g of AMPS (with a molar ratio of 0.5/3.0/1.0), and 0.5 g of modified lignin (5% wt. relative to the monomers) were dissolved in 16 mL of doubly distilled water and placed in a glass reactor. The solution was transferred to test tubes, and the redox initiators at 1% were added and stirred at 900 rpm under a N_2_ atmosphere during the polymerization reaction to displace the oxygen present in order to prevent it from interacting with the free radicals released by the redox initiator. Afterwards, the reaction medium was maintained at a temperature of 50 °C for 4 h and then extracted and dried at 50 °C in an oven for 7 days. The moisture-free hydrogel was swollen in water to remove the residual monomers or reactants and produce a pH of 5.6. Subsequently, it was dried to total dehydration at 50 °C for another 7 days. Figure 2 illustrates the potential hydrogel structure that was obtained, noting the functional chemical groups in the polymer chains.

The xerogel (moisture-free hydrogel) was crushed in a mortar and sieved to obtain a powder with a particle size between 600 and 125 µm, to maximize the contact surface between the bacterial strains and the hydrogel and thereby facilitate the diffusion of nutrients.

### 2.4. Bacterial Strains and Culture Conditions

*P. putida* and *B. cereus* were selected because they are lignin-degrading bacteria [17,18]. In addition, *B. paramycoides*, a *Bacillus* of the *B. cereus* group that was recovered in previous biodegradation tests and molecularly identified, was added. The biodegradability of the synthesized hydrogel was assessed as a function of the growth of the selected strains in a culture broth with the powdered hydrogel. Axenic cultures of *P. putida F1* ATCC−700007 were acquired from the American Type Culture Collection (ATCC), and *Bacillus cereus* and *Bacillus paramycoides* strains were previously isolated and identified at the University of Guadalajara. These strains originated from samples of water and soil contaminated with oil hydrocarbons, which were collected from the city of Guadalajara, México.

Before the evaluation of the growth of the bacteria in the presence of the hydrogel, each strain was independently cooled in test tubes with LB broth supplemented with glucose at a concentration of 2% *w*/*v*. For this procedure, three isolated colonies of each strain were taken and inoculated into 30 mL of LB, with incubation conditions of 35 ± 2 °C for 24 h without shaking. To verify the purity of the strains, each bacterium from the pre-inoculums was seeded separately using the quadrant streak plate method on Müller–Hilton agar. The incubation conditions were 35 ± 2 °C for 48 h. From the above growth, a bacterial consortium called the “bacterial pool” was formed by taking three isolated colonies of each strain and transferring them to a Falcon tube containing 20 mL of supplemented LB broth. The bacterial pool was incubated at 35 ± 2 °C for 24 h without shaking.

### 2.5. Biodegradation Experiments and Bacterial Growth

For the growth kinetics with the selected bacterial strains, 0.5 g samples of the hydrogel with a particle size between 600 and 125 µm were placed in 500 mL graduated screw flasks (ISOLAB, DMX-México) with 180 mL of LB broth (pH 7 ± 0.2) to function as the main source of carbon for the microorganisms. The flasks with the hydrogel and the LB broth were sterilized in an autoclave (15 min at 121 °C) and allowed to cool down to room temperature. Subsequently, they were inoculated with 20 mL of the bacterial pool. At the beginning of the bacterial treatment, an average bacterial concentration of log10 8.95 UFC/mL was obtained. Additionally, a positive control was prepared by mixing 175 mL of LB nutrient broth (pH 7 ± 0.2) with 5 mL of a 20% *w*/*v* glucose solution and 20 mL of the bacterial pool inoculum to compare the growth of the bacterial consortium under the same conditions but with glucose, which is an easily metabolically assimilated substrate for microorganisms. Furthermore, to observe if the LB broth and the culture conditions affected the hydrogel, a negative control was used in triplicate with the same media used in the kinetic experiments, but without the bacterial pool. The incubation conditions were 35 ± 2 °C for 60 days and without shaking. The experiments were carried out in triplicate.

The growth of the bacterial consortium in the presence of the hydrogel was monitored spectrophotometrically at an optical density (OD) of 600 nm (Jenway 6300 Spectrophotometer-Cole, Palmer, IL, USA). An aliquot of 1 mL was taken from each inoculated flask and the positive and negative controls with a periodicity of no more than 8 days to were used to indirectly determine the increase in biomass in terms of the increased absorbance of the culture. All experiments were carried out in triplicate.

### 2.6. Loss of Polymer Mass through Hydrogel Degradation by Bacterial Pool

After 60 days, the hydrogel was removed and washed with HPLC-grade water and dried in an oven at 50 °C for 2 days. The mass loss (as a percentage) was calculated using Equation (1) [8]:(1)$$W_{L}\%=\frac{W_{0}-W_{F}}{W_{0}}\left(100\%\right)$$
where *W_F_* is the final mass of the hydrogel at the end of the growth kinetics, *W_O_* is the mass of the hydrogel before the growth kinetics, and *W_L_* is the percentage of lost mass.

### 2.7. FTIR Spectra to Monitor Changes in Hydrogel Bonds after Bacterial Treatment

Fourier-transform infrared spectroscopy (FTIR) was used to record the possible chemical changes in the hydrogel due to the metabolic action of the microorganisms that were used. After being synthesized, sieved, and subjected to a negative control test (the hydrogel did not show chemical changes when exposed to the LB culture broth), the hydrogel was washed with HPLC-grade water and then dehydrated in an oven at 50 °C for 48 h. Its spectrum was evaluated to verify that its chemical structure remained unchanged. The resulting hydrogel after the growth kinetics with the microorganisms was processed in the same way as the control hydrogel. All spectra were collected at a resolution of 4 cm^−1^ in the range of 600–4000 cm^−1^ with a total of 32 scans using a NICOLET iS50 FTIR spectrometer (Thermo-Fischer Scientific Inc., Madison, WI, USA) equipped with an ATR sampling unit (at a temperature of 25 °C).

### 2.8. Scanning Electron Microscopy (SEM) to Evaluate Changes in Hydrogel Morphology

To evaluate the morphological surface changes in the hydrogel by the bacteria, the materials were characterized using a field-emission scanning electron microscope (FE-SEM) Tescan Bruker XFlash MIRA LMU model (TESCAN Co., Warrendale, PA, USA). The hydrogel samples before and after the growth kinetics were dehydrated in a lyophilizer (10 N, Ningbo Scientz Biotechnology Co., Ltd. Ningbo, Zhejiang, China) under a residual pressure of 5 kPa at −50 °C for approximately 48 h until obtaining a constant weight. The lyophilized hydrogel samples were immediately coated with a gold layer for 30 s at 1 mA, and after this treatment, images were obtained via SEM using an acceleration voltage of 20 kV.

### 2.9. Thermogravimetric Analysis

A TGA-DSC Discovery calorimeter obtained from TA Instruments (New Castle, DE, USA) was used to measure the rate of mass loss as a function of temperature in the hydrogel samples before and after degradation. The system was calibrated for temperature by analyzing the Curie temperatures of the reference materials, alumel (T = 425.75 K) and nickel (T = 631.35 K), with a heating rate of 10 K min^−1^. The calibration for mass was performed with the NIST standard masses of 1 mg and 100 mg.

### 2.10. Statistics

The data obtained during the growth kinetics were expressed in terms of the optical density at 600 nm (OD_600_) versus the incubation time (in days) and defined as means and standard deviation. In addition, increases in bacterial growth were calculated and compared with the kinetics control test (with glucose). The data were processed using the software Origin^®^ Pro 2016 (OriginLab Corporation, Northampton, MA, USA).

### 2.11. Elemental Analysis

Tests for the elemental analysis of the hydrogel before and after bacterial treatment were performed using a LECO Elemental Analyzer truspec micro (St. Joseph, MI, USA), whereby 2 mg of each sample was burned at 1273.15 K to obtain the weight percentage composition of C, H, N, and S. The elemental analyzer was calibrated with 10 tests using a sulfametazine standard from LECO (51.8% C, 5.1% H, 11.5% O, 20.1% N, and 11.5% S).

## 3. Results

### 3.1. Growth Kinetics of Bacterial Strains with Hydrogel

The growth of the bacterial strains was measured indirectly by determining the optical density of UV at 600 cm^−1^ (OD_600_). The growth behavior of BH1–3 and the control (C), as well as BH (the average of the three experiments with the hydrogel), is shown in Figure 3a,b. The difference in the results of the OD of the control bottle (C) is notable compared with the BH sample (*p* < 0.05). An exponential growth phase can be observed during the first 15 days. The growth of C showed its maximum reading on day 19 (OD_600_ = 5.184), while the BH1–3 flasks maintained constant growth without great differences in the spectrophotometrically measured readings. From day 20, the OD of C always remained higher than the readings for BH1–3.

However, the variations in the measurements were minimal, indicating that the culture in flask C remained in a stationary growth phase until the end of the kinetics with an average growth of OD_600_ = 3.00 ± 1.84. On the other hand, the growth kinetics of BH1–3 showed a slight increase on day 40 and remained constant until 60 days of recording, showing little difference between the readings (*p* < 0.35), with an average growth of OD_600_ = 1.09 ± 0.37. Pearson’s correlation coefficient for the BH1–3 kinetics was r = 0.86 (Figure 4). Figure 4 shows the statistical fit of all optical density measurements (blue squares) and the internal confidence limits (gray lines) show the global behavior with a confidence level of 0.95 for the degradation kinetics as a function of time. 

### 3.2. Hydrogel Mass Loss at the End of Growth Kinetics with the Bacterial Pool

At the end of the growth kinetics, the culture flasks were washed, and the hydrogels were recovered individually. The initial masses of the dried hydrogels were measured before they were subjected to bacterial degradation. The initial average weight of the hydrogels at the beginning of the kinetics was 0.50 g (*W*_0_), and at the end of the kinetics it was 0.42 g (*W_F_*), with an average mass loss of 18.4% (*W_L_%*).

### 3.3. FTIR Spectra Comparison

The FTIR spectra of the hydrogels were compared before and after the degradation treatment with the bacteria. In Figure 5, the spectra that were obtained are compared. Spectrum (a) corresponds to the control (named hydrogel before biodegradation (HBB)), wherein the peak at 2930 cm^−1^ is assigned to the O–H stretching vibrations [9,10] in the hydrogel, the peak at 1705 cm^−1^ corresponds to the C=O stretching of the carboxylic acid groups [11], while the peaks at 1647 cm^−1^ and 1548 cm^−1^ represent the stretching vibrations of the C=O and C–N groups of the amide groups, respectively [19,20,21,22]. The peak at 1447 cm^−1^ represents a scissor reflection of the CH_2_ group [19,21,23]. The peak at 1390 cm^−1^ corresponds to the CH stretching of the methyl group, the peak at 1150 cm^−1^ corresponds to the S=O stretching of the sulfite group [9,12], while the peak at 1034 cm^−1^ corresponds to the C–O stretching of the acid groups [12]. The peaks shown in the spectrum of the negative control correspond to the typical absorption bands of a polymer of this nature [19,20,21,22,23,24,25]. In contrast with the FTIR spectrum of the hydrogels recovered from BH1–3 (identified as hydrogel after biodegradation (HAB)) (see Figure 5b), the disappearance of the peak at 1705 cm^−1^ is notorious, showing an evident decrease in the absorption bands belonging to the carboxylic groups (at 2930, 1705, and 1647 cm^−1^).

### 3.4. SEM Results

The morphological changes in the control hydrogel (HBB) and recovered hydrogels (HABs) from the BH1–3 experiments were compared using SEM. The micrographs of the hydrogel before contact with the bacterial culture (HBB) are shown in Figure 6a–c (from 100 to 20 µm, respectively), and those of the hydrogels recovered after bacterial treatment (HABs) are shown in Figure 6d–f (from 100 to 20 µm, respectively).

Figure 7 shows photographs indicating the modification in the swelling behavior of the polymeric material before and after bacterial treatment. These images show that the modifications in the surface characteristics were caused by the action of the bacterial treatment.

### 3.5. Elemental Analysis Results

The elemental compositions of the hydrogels in terms of the weight percentages of *C*, *H*, *N*, and *S* before and after bacterial treatment were studied using elemental analysis. Table 1 illustrates the results that were obtained with this technique. This analysis was used to corroborate the degradation of the samples by the bacterial consortium to achieve the main objective of the “mixture of bacteria” degrading the polymeric material. We observed that all the elements were lost. Because the system used oxygen to oxidize the samples, the results do not report the amounts of oxygen; however, it is possible to estimate the amounts of oxygen in the samples via the difference in amounts before and after degradation. The amount of oxygen in the control sample before degradation was 57.73%, while it was 54.41% after degradation, indicating a difference of 3.32%. The loss of all the elements was approximately 15.46% (see Table 1), which is a reference value and comparable with the percentage obtained from the gravimetry measurements of 18.4%. Furthermore, the differences in the amounts of carbon and nitrogen are appreciable since the bacteria metabolized these elements.

### 3.6. TGA Results

The thermogravimetric analysis (TGA) of the samples is depicted in Figure 8. The thermal degradation temperatures of the samples were assessed to confirm the degradation of the hydrogel before (HBB) and after bacterial treatment (HAB). In the figure, it is possible to observe the different thermal behaviors of both samples. The TGA curve for the original hydrogel (HBB) shows that thermal decomposition started at 200 °C, which was attributed to functional groups containing oxygen (i.e., carboxylic acid and amides). These groups in the polymer (hydrogel) come from lignin (the crosslinking agent) and acrylamide and acrylic acid (monomers). Using TGA, Kanishka K. et al. [26] have demonstrated that functional groups with oxygen start to degrade at temperature close to 200 °C. Our results suggest that the original hydrogel (HBB) contained a greater amount of these functional groups, confirming that the number of these functional groups decreased after bacterial treatment due to the degradation carried out by the bacterial consortium. Our results are consistent with those reported by Safaa et al. [27] for the decomposition of the last functional groups (amines) at the temperatures recorded herein. The weight loss of HBB was 1.6% between 100 and 200 °C, and 14.8% between 200 and 400 °C, and total decomposition occurred at 530 °C. For HAB, the weight loss was 0.03% in the interval of 100–200 °C and 3.31% between 200 and 400 °C, corroborating that the loss of the CHCOOH and CHCONH2 groups occurred between 200 and 400 °C. In addition, the mass losses between these temperatures are consistent with the results from the gravimetric and elemental analyses.

## 4. Discussion

The growth variability, in terms of the standard deviation, was greater in the hydrogel cultures than in the control culture. This may be because even if a powder with an established range is obtained, the contact surface between the bacteria and the particles may vary, and the composition and availability of the substrate do not present the most favorable conditions for bacterial growth. This is due to the improved access to the carbon and nitrogen elements of the hydrogel, which is necessary for the microorganisms to activate the enzymatic systems and adapt to the new conditions, leading to energy being spent that could affect the rate of bacterial growth. The weight loss after bacterial treatment was 18.4% wt, indicating that the bacteria were able to assimilate the other hydrogel compounds as substrates in addition to the lignin, and join one or more metabolic pathways, which was demonstrated in the ability of the bacteria to maintain their viability and cultivability that was monitored throughout the kinetic treatment. Potential compounds that were used after lignin were acrylamide and acrylic acid since it has been reported that some microorganisms can grow with acrylamide and have been proposed as possible candidates for bioremediation, including bacteria such as *Pseudomonas* and *Bacillus* [6]. With experiments on the degradation of hydrogels with lignin carried out on soils, Song et al. [13] obtained a mass loss of 6% in the first 60 days, which increased to 14% in 120 days. The mass losses in our results are superior because the experiments were carried out in the liquid phase. However, the degradation of AM/AAc /AMPS hydrogels with lignin may be slower in contaminated soils or, as shown in the results of Mittal et al. [28] for AM/AAc hydrogels with ghatti gum, this degradation may reach up to 89% if the soil is compost soil or rich in microorganisms.

The FTIR experiments demonstrated that the carboxylic groups present in the lignin, acrylamide, and acrylic acid of the hydrogel decreased during bacterial growth. This indicates that the bacterial consortium had a preference for these components of the hydrogel. Furthermore, this effect on the material was confirmed by the superficial morphological changes that were highlighted with SEM, such as the sheet thinning in the structure of the HAB, as well as the absence of granular protrusions on the surfaces of the sheets. However, the structure of the three-dimensional network and the size of the pores did not disappear, which, according to Wang et al. [4] and Song et al. [13], are important characteristics of water absorption. This is shown more clearly in the images in Figure 6a,d, in which the morphological and structural changes in the recovered hydrogels from BH1–3 are evident.

Based on the results, it is evident that the microscopic structure of the hydrogel changed. These microscopic changes modified the swelling of the hydrogel. The water absorption capacity of the hydrogel was evaluated after bacterial treatment. Pulverized xerogels (which were previously washed and dried) without treatment (HBB) and the xerogels that were recovered at the end of the bacterial growth kinetics (HAB) were placed directly into 500 mL of distilled water for more than 72 h to determine their maximum swelling capacities. A swelling of 7940% was obtained from the dried xerogel without treatment (HBB, Figure 7a) and the same sample when it was subjected to maximum swelling (Figure 7b). Similarly, Figure 7c shows a micrograph of the xerogel after bacterial treatment (HAB), and Figure 7d after swelling, obtaining a swelling of 43,080%. Swelling after treatment was higher in hydrogels that underwent biodegradation. This may be because interactions between the polymeric chains were lost, that is, certain physical and chemical crosslinks were lost because parts of the polymer chains were “consumed” by the bacteria; nevertheless, there was still a degree of crosslinking. For these reasons, swelling continued to exist despite the pore size (as observed with SEM) being preserved. These results can be interpreted in two ways: the first is that the partially synthetic hydrogel was assimilated by bacteria that are regularly found in soils of temperate climates (since the bacteria used in this project belong to the mesophilic range), and the second is that the water retention capacity of the hydrogel was favorable, suggesting that this type of hydrogel has the potential to be used to capture pollutants from soils in which a native microbiota is present, whereby the hydrogel will continue to retain its absorbent capacity despite the occurrence of partial biodegradation.

## 5. Conclusions

The bacterial consortium consisting of *Pseudomonas putida*, *Bacillus cereus*, and *Bacillus paramycoides* was able to use the hydrogel as the main carbon source and degrade this polymeric material by approximately 18.4% wt within a period of 60 days at 35 ± 2 °C.

With the tests performed on the hydrogel after bacterial treatment, we observed the bacterial consortium’s potential preference to assimilate carboxylic acid and acrylamide functional chemical groups, and the morphology of the hydrogel was affected. Furthermore, the hydrogel’s swelling capacity was modified due to the loss of factors from its polymeric chains. To confirm the weight loss, an elemental analysis (EA) was performed, obtaining a value of 15.46%, and the thermogravimetric analysis (TGA) demonstrated that the bacterial consortium degraded the carboxylic and amide functional groups.

We conclude that the bacteria were able to use the hydrogel as their main carbon source, although the metabolic incorporation of the substrate was not simple. Hydrogels are cross-linked polymers that are difficult to degrade before decomposition. In this study, degrading part of the hydrogel was made possible by taking advantage of the bacteria’s preference for lignin, due to which they were able to degrade the polymer chains.

## Figures and Tables

**Figure 1 polymers-15-02588-f001:**
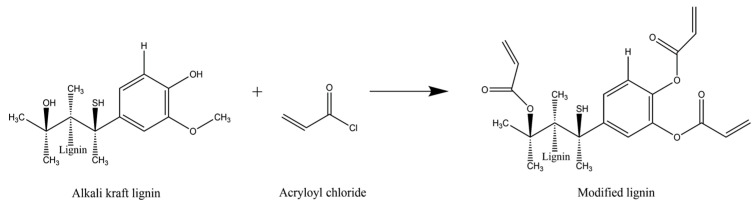
Proposed reaction scheme for the modified lignin used as crosslinker.

**Figure 2 polymers-15-02588-f002:**
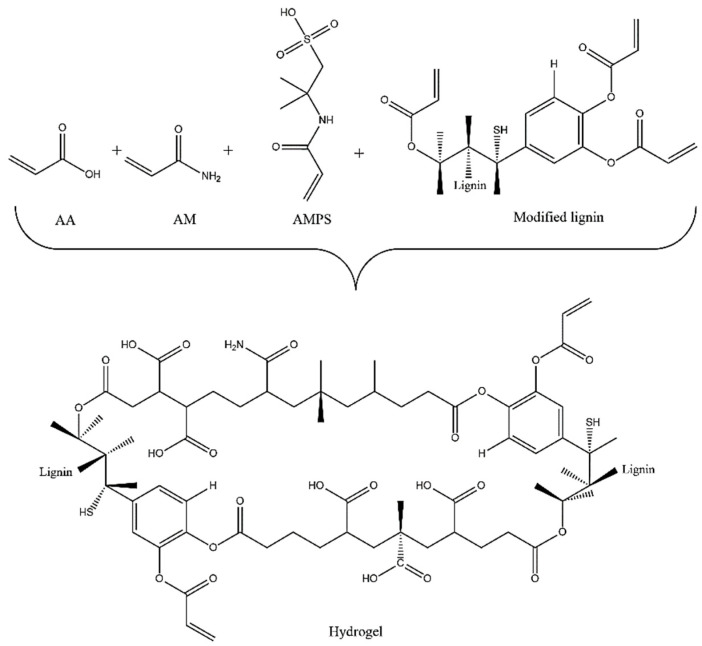
Potential structure of hydrogel that was obtained.

**Figure 3 polymers-15-02588-f003:**
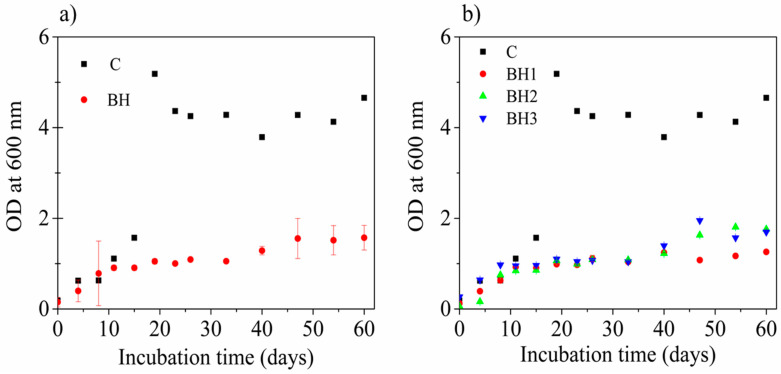
(**a**) Bacterial growth kinetics (bacterial pool) in control flask with glucose (C) and culture broth average (BH) with powdered hydrogel for 60 days. (**b**) C growth kinetics and cultures with hydrogel (BH1–3).

**Figure 4 polymers-15-02588-f004:**
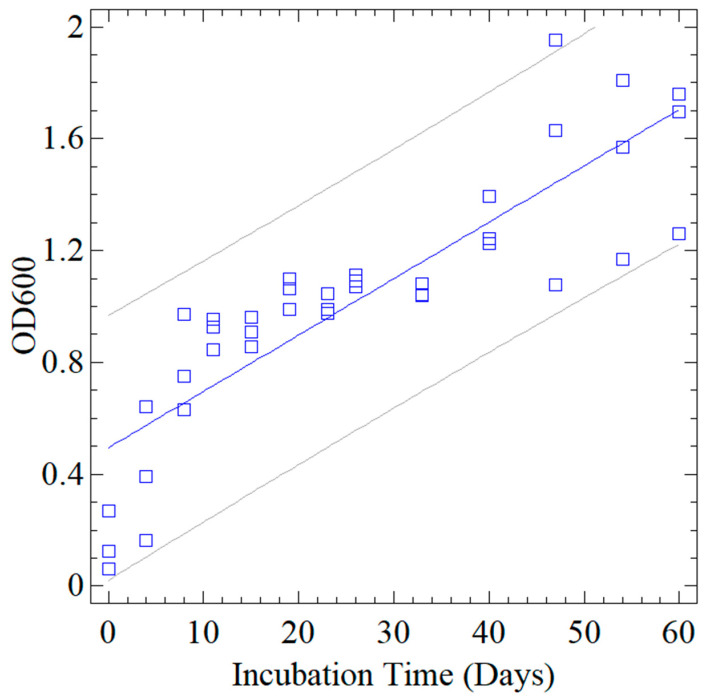
Graph of fitted model with the confidence limits (95%) and prediction limits for the BH1-3 kinetics.

**Figure 5 polymers-15-02588-f005:**
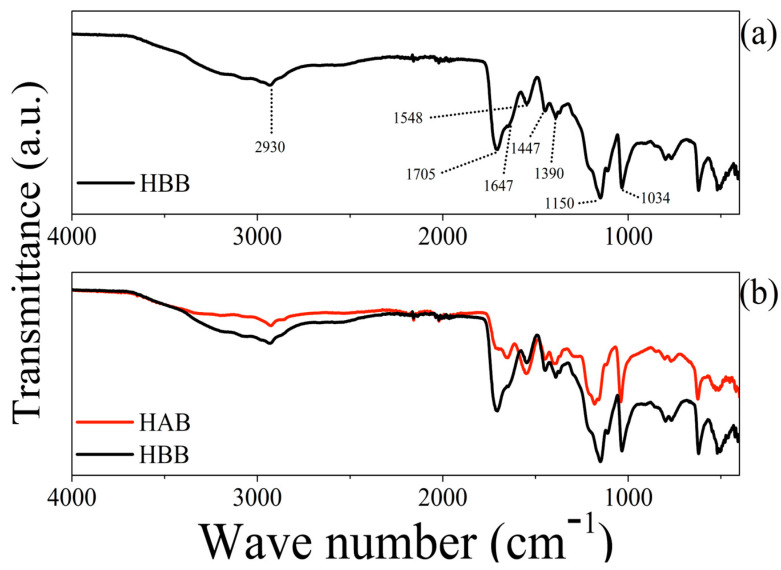
FTIR spectra of (**a**) hydrogel before biodegradation (HBB) and (**b**) hydrogel after biodegradation (HAB). Note that peak at 1705 cm^−1^ is missing after growth kinetics.

**Figure 6 polymers-15-02588-f006:**
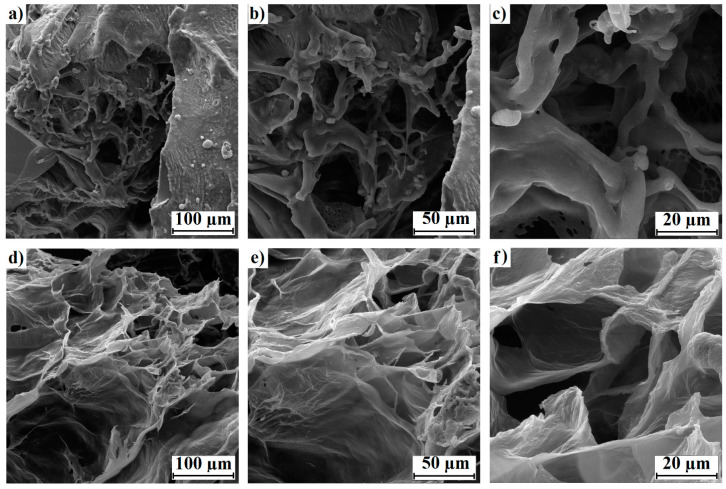
Morphologies using SEM of (**a**) HBB at 100 µm, (**b**) HBB at 50 µm, (**c**) HBB at 20, (**d**) HAB at 100 µm, (**e**) HAB at 50 µm, and (**f**) HAB at 20 µm.

**Figure 7 polymers-15-02588-f007:**
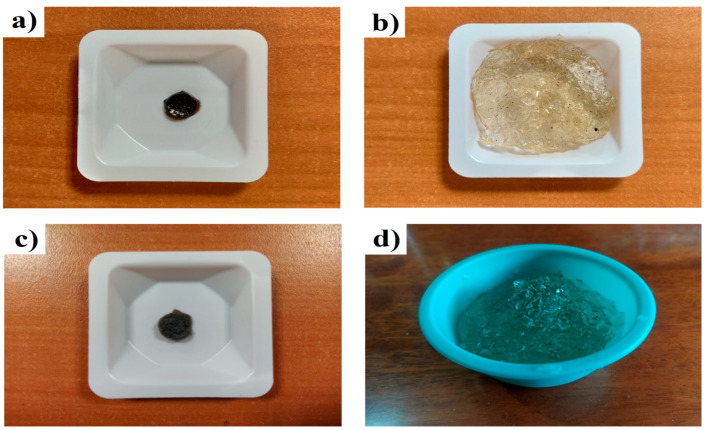
Photographs of swelling for (**a**) xerogel HBB, (**b**) HBB at maximum swelling, (**c**) xerogel HAB, and (**d**) HAB at maximum swelling.

**Figure 8 polymers-15-02588-f008:**
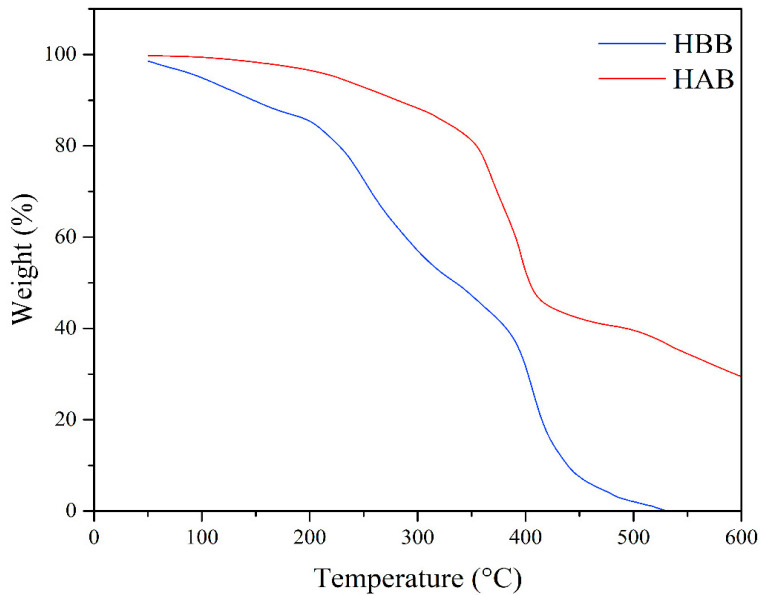
Comparison of the thermal behaviors of HBB and HAB via thermogravimetric analysis (TGA).

**Table 1 polymers-15-02588-t001:** Results of elemental analysis.

*Sample*	*% C*	*% H*	*% N*	*% S*
Control hydrogel (HBB)	35.01	4.98	11.48	6.26
Degraded hydrogel (HAB)	29.56	4.71	5.96	5.36
Weight difference	5.45	0.27	5.52	0.9

## Data Availability

Not applicable.

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
