# Peer review of "Study on the Degradation of a Semi-Synthetic Lignin–Acrylic Acid Hydrogel with Common Bacteria Found in Natural Attenuation Processes"

_polymers, 2023, doi:10.3390/polym15122588_

Round 1

Reviewer 1 Report

Thank you for submitting your paper to Polymer journal. I read your article and wanted to provide some feedback that may help improve your writing.

Firstly, I found your language to be quite complex and at times difficult to understand. I suggest using simpler language and avoiding convoluted sentence structures to make your writing more accessible to a wider audience.

Additionally, I noticed that you haven’t reported any mechanical tests for the synthesized hydrogel. Please add compression data.

it must be improved 

Author Response

REviewer  1

Comments and Suggestions for Authors

Thank you for submitting your paper to Polymer journal. I read your article and wanted to provide some feedback that may help improve your writing.

Observation 1:

Firstly, I found your language to be quite complex and at times difficult to understand. I suggest using simpler language and avoiding convoluted sentence structures to make your writing more accessible to a wider audience.

Answer 1:

Comprehensive language review was performed through the service provided by the MDPI system.Several changes were made.

Observation 2:

Additionally, I noticed that you havent reported any mechanical tests for the synthesized hydrogel. Please add compression data.

Answer 2:

In a subsequent work, the mechanical properties of hardness, compression and flexion of these hydrogels will be reported. I will show you the hardness (Shore D) of the hydrogels before and after the treatment as a preliminary result, but it is important to note that this research was focused on sustainability of the hydrogels than properties of hydrogel as a mechanism to remove pollutants in water.

Shore hardness measurements were performed in triplicate.

For HBB (Hydrogel before biodegradation)—- 32.8 HD

For HAB (Hydrogel after biodegradation)——- 23.1 HD

which is equivalent to a decrease in hardness of 29.5%

Reviewer 2 Report

 The manuscript aims to study the biodegradation of a hydrogel crosslinked with chemically modified lignin by a consortium of bacteria. The modified lignin promotes the hydrogel's degradation, and the hydrogel serves as a source of carbon and nitrogen for the bacteria.

The work's originality lies in using modified lignin as a cross-linker. The study also evaluates the ability of soil microorganisms to use hydrogel as a carbon source, which has not been extensively studied before.

I have a few comments: 

The final paragraph of the introduction should be rewritten to clearly show the novel activities that have been undertaken in this study (avoid putting state-of-the-art in your last paragraph of the introduction)

The N2 atmosphere is not supposed to prevent the formation of free radicals but to protect them from O2 interference. Revise your statement: “…under N2 atmosphere during the polymerization reaction to avoid free radicals formation…”

I believe the authors should show the speed of the hydrogel consumption by the pool of bacteria when the hydrogel is in bulk form and not crushed. This way, they can propose a real-world scenario for their study. Moreover, I suggest authors, in their future work, study the kinetic of hydrogel formation by different weight ratios of lignin.  

A few minor issues with phrasing and punctuation could be improved. For example, the following sentence appears broken: “In this work, the pool of bacteria and not individual bacteria, since several authors have reported that plastic biodegradation tests using individual bacteria are less efficient.” Or this sentence: “We aimed to the hydrogel could be used by the bacterial metabolic systems as a carbon source.” Or there is a missing “and” in here: “modified lignin (5% wt. relative to monomers) was dissolved in 16 mL of doubly distilled water were placed in a glass reactor.”

I suggest authors proofread the manuscript carefully.

Author Response

Reviewer 2

Comments and Suggestions for Authors

 The manuscript aims to study the biodegradation of a hydrogel crosslinked with chemically modified lignin by a consortium of bacteria. The modified lignin promotes the hydrogel's degradation, and the hydrogel serves as a source of carbon and nitrogen for the bacteria.

The work's originality lies in using modified lignin as a cross-linker. The study also evaluates the ability of soil microorganisms to use hydrogel as a carbon source, which has not been extensively studied before.

I have a few comments: 

Observation 1:

The final paragraph of the introduction should be rewritten to clearly show the novel activities that have been undertaken in this study (avoid putting state-of-the-art in your last paragraph of the introduction)

Answer 1:

Last two paragraphs was changed and rewritten, the state of the art was realized due to an earlier remark by the editor.

Changes realized:

In this study a pool of bacteria rather than individual bacteria in this work since several authors have reported that plastic biodegradation tests using individual bacteria are less efficient. Shovitri et al. [14] used Pseudomonas and Bacillus isolated from mangrove waters to degrade single-use plastics using the Winogradsky column method. Then, they demonstrated that the bacteria enacted a bioaugmentation process in marginal soils contaminated with plastics, indicating a positive effect on degradation. However, by using these microorganisms separately, they discussed the effects of the particular enzymes of each microorganism on the plastics, obtaining different results from when the bacteria were used in a consortium. On the other hand, Aravinthan et al. [15] studied the synergistic growth of a P. azotoformans and Bacillus flexus consortium, obtaining a 22.7% degradation of physically pretreated polypropylene. In previous studies by these authors, it was reported that the effect of using a single bacterium for biodegradation is lower than when using a consortium.

For the above reasons, in this study was evaluated the modified hydrogel's ability to maintain the viability of P. putida F1 and two types of Bacilli, B. cereus and B. paramycoides, in a bacterial pool where the hydrogel functioned as a source of carbon and nitrogen for these microorganisms. The bacteria were not used individually because the degradation of complex materials is more efficient when microorganisms are used in a consortium. The bacterial growth kinetics were evaluated at a controlled temperature of 35 ± 0.2°C, without shaking. The hydrogel's structural changes, mass loss, and final composition were evaluated to analyze its degradation via the degradation of its lignin content and to demonstrate that the bacteria were able to use the carbon chains of the synthetic material as a carbon source.

Observation 2:

The N2 atmosphere is not supposed to prevent the formation of free radicals but to protect them from O2 interference. Revise your statement: “…under N2 atmosphere during the polymerization reaction to avoid free radicals formation…”

 Answer 2:

Nitrogen was used to displace the oxygen present to prevent it from interacting with the free radicals released by the redox initiate. Text was changed.

Observation 3:

I believe the authors should show the speed of the hydrogel consumption by the pool of bacteria when the hydrogel is in bulk form and not crushed. This way, they can propose a real-world scenario for their study. Moreover, I suggest authors, in their future work, study the kinetic of hydrogel formation by different weight ratios of lignin.  

Answer 3:

The objective of crushing the hydrogel is for the pool of bacteria to interact better with the material, since in preliminary tests it was used in bulk, and although it was also degraded, only the time for degradation was affected. The degradation time reported here is 60-70 days, in bulk it is almost double the time, but the important analysis was the degradation of these cross-linked polymers.

As a continuation of the project, it is intended to analyze how degradation is affected depending on the amount of lignin (3%, 8%, 11% and 15%). That study will be done. The suggestion is appreciated.

Observation 4:

Comments on the Quality of English Language

A few minor issues with phrasing and punctuation could be improved. For example, the following sentence appears broken: In this work, the pool of bacteria and not individual bacteria, since several authors have reported that plastic biodegradation tests using individual bacteria are less efficient.” Or this sentence: We aimed to the hydrogel could be used by the bacterial metabolic systems as a carbon source.” Or there is a missing and” in here: modified lignin (5% wt. relative to monomers) was dissolved in 16 mL of doubly distilled water were placed in a glass reactor.”

I suggest authors proofread the manuscript carefully.

Answer 4:

Comprehensive language review was performed through the service provided by the MDPI system.Several changes were made.

Round 2

Reviewer 1 Report

I wanted to reach out and inform you that I have completed the review of your revised manuscript, Study of the degradation of a semi-synthetic Lignin-Acrylic acid hydrogel by common bacteria founded in natural attenuation processes. I am pleased to let you know that the revisions you made have addressed the concerns and suggestions raised during the review process. Based on the thorough revisions made, I am confident in recommending your manuscript for publication.